# The Radial Growth of *Juniperus squamata* Showed Sharp Increase in Response to Climate Warming on the Three-River Headwaters Region of Tibetan Plateau since the Early 21st Century

**Guoqing Zhao [1,2], Zhongbao Xin [1,2,*], Jinhao Liu [1,2], Yanzhang Huang [3], Maierdang Keyimu [4] and Zongshan Li [3]**

1   College of Water and Soil Conservation, Beijing Forestry University, Beijing 100083, China; zhao722529@bjfu.edu.cn (G.Z.); hawkey98@126.com (J.L.)
2   Ji County Station, Chinese National Ecosystem Research Network (CNERN), Beijing Forestry University, Beijing 100083, China
3   State Key Laboratory of Urban and Regional Ecology, Research Center for Eco-Environmental Sciences, Chinese Academy of Sciences, Beijing 100085, China; yzhuang_st@rcees.ac.cn (Y.H.); zsli_st@rcees.ac.cn (Z.L.)
4   State Key Laboratory of Desert and Oasis Ecology, Xinjiang Institute of Ecology and Geography, Chinese Academy of Sciences, Urumqi 830011, China; mhk@ms.xjb.ac.cn
*   Correspondence: xinzhongbao@126.com

**Abstract:** In order to explore the impact of climate change on the ecosystem at high altitudes, dendroclimatology was used to study the response of radial growth of *Juniperus squamata* Buch.-Ham. ex D.Don to the rapid warming in Nangqian County over the past 60 years, and a tree-ring width chronology for 115 years was established. (1) Meteorological data showed that the temperature in Nangqian County of the Tibetan Plateau has increased continuously during the past 60 years, and the minimum temperature has had the most significant change (0.63 °C/10a), especially between 2000–2019 (0.12 °C/a). Over the same time period precipitation has not changed significantly (0.94 mm/a, $p > 0.10$). The standard chronology was used to reconstruct the mean temperature series from July to September in Nangqian meteorological station during the past 115 years (1905–2019). The explained variance of the reconstructed equation was 42.8% (40.8%, after adjusting for degrees of freedom). The reconstructed temperature series can be roughly divided into two stages: from 1905 to 1999, the temperature fluctuated around the average value, 12.10 °C, and from 2000 to 2019, the temperature showed a significant upward trend. (2) The analysis of the climate-tree growth relationship indicated that the response of radial growth of *Juniperus squamata* to temperature was significantly stronger than the response to precipitation; especially in the last 20 years, when the radial growth of *Juniperus squamata* was positively correlated with temperature ($p < 0.01$). Compared to the maximum temperature and mean temperature, the correlation between radial growth of *Juniperus squamata* and minimum temperature was more significant. (3) Under the background of climate warming, the radial growth trend of *Juniperus squamata* in Nangqian county was consistent with temperature changes. Particularly in the past 20 years, the radial growth of *Juniperus squamata* showed a significantly increased trend and entered a rapid growth period.

**Keywords:** Tibetan Plateau; *Juniperus squamata*; tree ring; dendroclimatology; temperature variations

## 1. Introduction

As the highest plateau in the world, the Tibetan Plateau (TP) is an important barrier for China's ecological and national security, with an area of about 2.5 million km² and an average altitude of more than 4000 m [1–4]. It has a profound influence on China's climate background. The TP significantly affects the Asian monsoon system and northern hemisphere atmospheric circulation due to its unique topography and dynamic and

thermal effects [5–8]. It is also the source of major rivers in China, such as the Yangtze River, Yellow River, and Lancang River, and is a key area for the global energy and water cycle [9]. Furthermore, the TP is also the initiator, amplifier, and regulator of global climate change [8,10]. Therefore, it is of great significance to study the climate change of TP in the historical period. However, due to the scarcity of meteorological stations and observation data, the study of TP climate change and its cause based on instrumental data is limited, and it is insufficient to understand the whole picture of TP climate change.

Tree rings have the advantages of accurate dating, good continuity, and high resolution when they are used to reconstruct past climatic factors in a certain area according to climate-tree growth relationship analysis [11–16]. Numerous studies have obtained complete historical temperature and precipitation data based on tree ring width (TRW) data, which lays a foundation for future predictions [17]. For example, Cufar [18] and Büntgen [19] used oak trees to reconstruct summer meteorological data; D'arrigo [20], Buckley [21], and Youngblut [22] reconstructed the temperature data of North America for more than 300 years with TRW data. Linderholm et al. [23] reconstructed temperature and precipitation data from 1786 to 2000 using *Pinus sylvestris* L. Buckley et al. [24] established a TRW chronology in northern Thailand and studied regional dry-wet changes. Wilson et al. and Anchukaitis et al. reconstructed the last millennium's summer temperature data with tree rings in the Northern Hemisphere [25,26]. Long time-scale climate change research is crucial to understand the climate characteristics and change patterns of the TP [27]. The reconstruction of historical average temperatures could enrich the understanding of regional thermal variation from a long-term perspective and from current warming trends, in addition to forest growth responses to such warming trends [15,16,28].

According to the fifth report of IPCC, global average surface temperatures have risen by 0.85 °C since 1880 [29]. Studies show that the temperature rise in the TP is about 30 years ahead of global warming; the average temperature increase in each decade is more than 0.3 °C, which is approximately twice the global average temperature increase over the same period, especially in winter [30–32]. Vegetation is the most active part of terrestrial ecosystem; it connects the atmosphere, hydrosphere, pedosphere, and biosphere. Tree growth is sensitive to climate change. Therefore, clarifying tree-climate dynamic response is helpful to evaluations of the impact of climate change on the forest ecosystem [33–36], to predictions of the change in the forest ecosystem carbon sink, and to sustainable forest management [37]. It is generally thought that warming will enhance growth of trees in cold regions, unless there is temperature-induced drought stress [38–41]. Numerous studies show that tree growth above the tree line is mainly limited by temperature, and the annual minimum temperature is the most significant indicator [42]. In alpine mountains, above the forest line, shrubs such as *Juniperus pingii* W. C. Cheng, *Hippophae rhamnoides* L., and *Rhododendron simsii* Planch., are controlled by the low temperature in the growing seasons [43]. Zhang et al. [44] used tree rings from Qamdo, eastern Tibet andfound that the mean minimum temperature was the most significant factor, and used tree rings to reconstruct the mean minimum temperature for the past 413 years. Gou et al. [45] found that in the source of the Yellow River, standard TRW chronology is significantly correlated to the minimum temperature. Zhu et al. [46] analyzed climate-*Juniperus przewalskii* Komarov. growth in Wulan area of Qinghai Province and found that the growth was limited by temperature. In the past decade, many ecologists have used changes in shrub growth and range as sensitive ecological indicators to reflect global climate change [47]. Compared to trees, shrubs distributed in alpine regions have better adaptability to alpine climates [43,48], and in most high-altitude regions, shrub species are more abundant than trees. The shrub distribution area is the concentrated population distribution area of the TP. In this paper, the radial growth process of *Juniperus squamata* and its correlation with climate factors, such as temperature and precipitation, were studied. Furthermore, attention was paid to the long-term change of the annual mean temperature in this area, to provide a scientific basis for the assessment of the suitability of shrub growth and human settlement on the TP, considering climate change effects, under the background of a warmer and more humid climate.

*Juniperus squamata* Buch.-Ham. ex D.Don is a shrub in the genus Cypress in the family Cypressidae, and is the representative cold-temperate needle-leaved forest in the eastern TP [49]. It has the characteristics of a deep root system, with strong stress resistance and long life span. *Juniperus squamata* are widely distributed in the source region of the Lancang River, China, and they play an important role in soil and water conservation, biodiversity conservation, and climate control. The objectives of the present study were to: (1) develop a TRW chronology of *Juniperus squamata* in the Three-River headwaters of the TP, and analyze its growth status, (2) determine the key climatic factors that limit radial growth of *Juniperus squamata* in the Three-River headwaters of the TP, and (3) reconstruct the mean temperature variability of the TP from July to September in the past 115 years in the context of climate warming.

## 2. Study Area

The sampling site is located at Nangqian County, Yushu Tibetan Autonomous Prefecture, Qinghai Province (Figure 1), with an average elevation of 4309 m. Located in the southeastern TP, Nangqian county is a transition zone from canyon area to the plateau. The landform is mountainous, high in the northwest and low in the southeast. The five major rivers, Baqu River, Ziqu River, Jiqu River, Requ River, and Zaqu River in Nangqian belong to the Lancang River system.

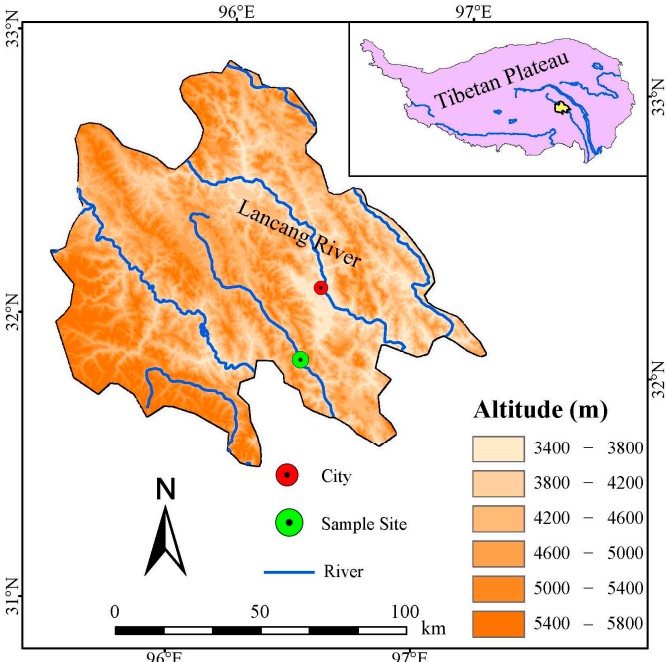

**Figure 1.** Location map of the tree-ring sample site and meteorological station in the Three-River Headwaters Region of the Tibetan Plateau.

The climate of the area is under the influence of the westerly and monsoon circulations of the Indian and Pacific Oceans, with pronounced dry and wet seasonality [14,50–52]. According to the climate data (time span 1957–2019) from meteorological stations (96°16′48″ E, 32°7′12″ N, 3643.7 m) close to tree-ring sampling sites, from 1957 to 2019, the extreme maximum and minimum temperatures were 28.1 °C and −24.1 °C, respectively (Figure 2). The mean annual temperature was 4.4 °C, the total multi-years' annual precipitation was 535.8 mm, and the mean annual relative humidity was 53% (Figure 3). The major shrub species in this area is *Juniperus squamata*; other shrub species mainly include *Dasiphora fruticose* L., *Hippophae rhamnoides* Linn., etc. [53]. The relatively steep topography at high altitudes in southeastern TP is not favorable for soil development; therefore, a thin soil

layer consists mainly of Alpine desert soil, Alpine meadow soil, Chernozem, Mountain meadow soil, Grey cinnamon soil, Alpine steppe soil, Chestnut soil [54].

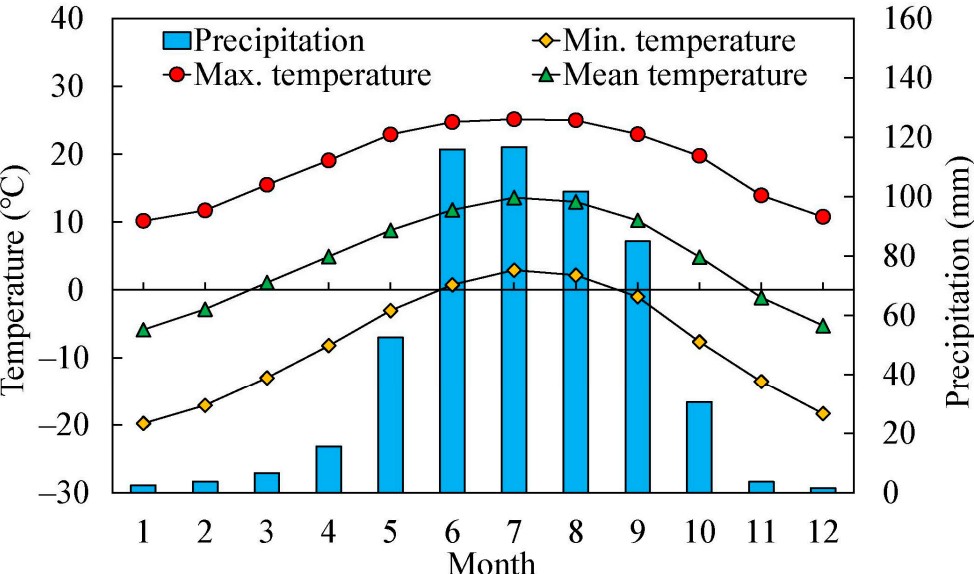

**Figure 2.** Seasonal distribution of temperature and precipitation in the Three-River Headwaters Region of the Tibetan Plateau during 1957 to 2019. Total monthly precipitation (blue bars), mean minimum temperature (yellow squares), mean maximum temperature (red circles) and mean temperature (green triangles).

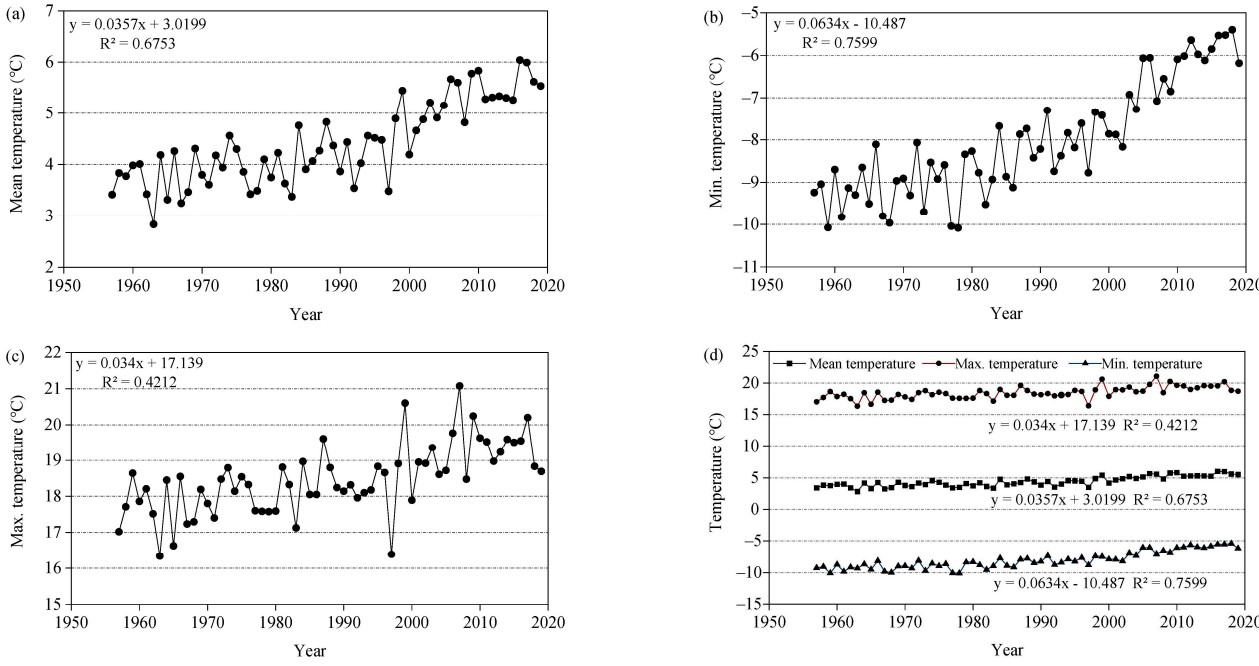

**Figure 3.** *Cont.*

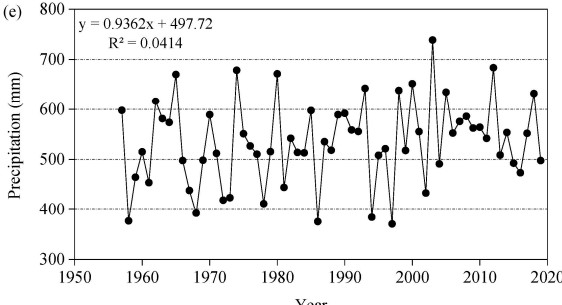
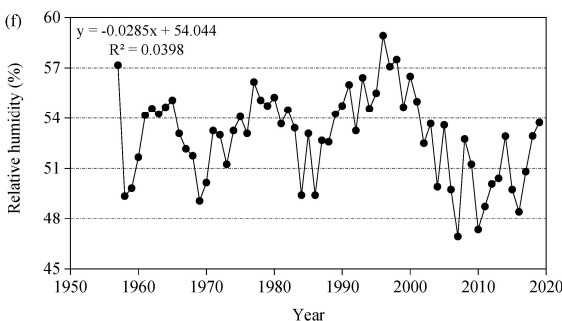

**Figure 3.** Climatic change trend in the Three-River Headwaters Region of the Tibetan Plateau from 1957 to 2019. (**a**) Mean temperature, Monthly mean temperature; (**b**) Min. temperature, Monthly mean minimum temperature; (**c**) Max. temperature, Monthly mean maximum temperature; (**d**) Temperature, Monthly mean maximum, mean, and minimum temperature; (**e**) Precipitation, Total annual precipitation; (**f**) Relative humidity, Monthly mean relative humidity.

## 3. Materials and Methods

### 3.1. Tree-Ring Data

In July 2019, we collected 20 *Juniperus squamata* discs in an area with little human interference, in Nangqian County (96°25′30″ E, 31°56′51″ N), on the southeastern TP. The samples were mainly selected from the main branches or side branches with healthy growth and strong stems, and the discs were intercepted along the base of the stem. The collected discs were preprocessed according to the international tree ring sample processing standard. In the laboratory, after the shrub discs were naturally dried, we polished them with P400, P800, and P1200 sandpapers, in turn, until the tracheary element, parenchyma, and ring boundaries of *Juniperus squamata* could be clearly seen under the microscope [51]. We took a radius in the east-west and north-south directions of the discs' facet, and the width of the rings was measured using WinDENDRO system with a resolution a 0.01 mm. The COFECHA software was utilized to check the results of cross-dating [52]. Finally, 40 radii of 20 discs were used for further study. A negative exponential curve or a linear regression with a negative slope was applied to remove the age-related trend for each series, using the ARSTAN program [50–52]. Tukey's biweight robust mean method was applied to average the dimensionless TRW indices, to achieve a standard chronology. The reliability and signal strength of standard chronology was assessed using a 50-year moving expressed population signal (EPS) with a 25-year overlap, and the mean series inter-correlations.

### 3.2. Climate Data

Monthly maximum, minimum, mean temperatures, and precipitation climate data were obtained from the Nangqian meteorological station (96°16′48″ E, 32°7′12″ N; 3643.7 m altitude), which was the station nearest to the tree ring sampling site (straight-line distance, 28.94 km). We obtained interval data from 1957 to 2019 (National Meteorological Science Data Center: http://data.cma.cn/, accessed on 12 December 2021).

### 3.3. Data Analysis

To determine the key climatic factors influencing radial growth of tree, we conducted growth-climate correlation analyses between TRW chronology and climate variables (maximum temperature, minimum temperature, mean temperature, and precipitation) using the correlation and response function analyses provided in the Dendroclim 2002 software. We used a bootstrap response function analysis with 1000 resampling, at the statistically significant 95% confidence level, for the 1957–2019 period [51]. Considering that tree growth is not only influenced by climatic conditions during the current year, but also from the previous year (legacy effect), our climate-tree growth relationship analyses included monthly windows from the previous year's June to the current year's October. A moving correlation analysis between TRW chronology and annual mean temperature was

conducted at 34-year intervals to investigate the temporal variability of the correlation between radial growth of *Juniperus squamata* and each limiting factor. A moving correlation analysis was also conducted between TRW chronology and precipitation to investigate the dynamic influences of precipitation on alpine shrub growth, and to test whether water stress influenced shrub growth.

*3.4. Climate Reconstruction and Testing Methods*

The precondition of tree-ring climate reconstruction is to determine the suitable meteorological factors, to establish the functional equations of the meteorological factor series and the chronological series through the correlation screening between radial growth and climate response analysis, and then, to extrapolate the region's historical climate. In tree-ring climatology, most of the models used in climate reconstruction are linear regression models, such as:

$$Clim = a \times crn + b.$$

In this model, *Clim* is the meteorological factor series, *crn* is the chronological series, and *a* and *b* are the constants. The constants are derived from the common time period between the least squares chronology and the climate factors. The corresponding chronology data representing the blank climate factor series with no data in the same time period are substituted into a function to obtain the reconstructed climate factor series. Adjusting the equation for the size of the equation $R^2$, allows the addition of independent variables to the right of the equation, such as chronological data from the previous year. The advantage of a multivariate linear equation is the ability to identify variables in a stepwise regression manner.

Due to the short recording period (1960–2019) at the Nangqian meteorological station, we used the "leave-one-out" calibration to test the stability of the reconstructed model. The specific implementation is described below.

In an overlapping period, eliminate a certain year, establish a regression equation, and then substitute the chronological value of the eliminated year into this equation to obtain estimated values of the year. Repeat the above steps until the estimated sequence of climatic elements is obtained, and compare against the measured values to test the stability [55]. In this study, the specific methods used, and parameters obtained for testing were the sign test, average value of product, reduced error, and correlation coefficient. The test results show if the equation is stable at the 0.05 significance level. Finally, compare the measured value with the reconstructed value and, accordingly, determine if the result is reliable.

## 4. Results

*4.1. Climate Change Trend*

The multi-year mean temperature of Nangqian from 1957 to 2019 was $4.43 \pm 1.61\,°C$, of which, the lowest temperature was $2.82\,°C$, recorded in 1963 (Figure 3a). The temperature showed an obvious increasing trend from 1957 to 2019 ($p < 0.001$). Minimum temperature, mean temperature, and maximum temperature increased at $0.63\,°C/10a$, $0.36\,°C/10a$, and $0.34\,°C/10a$ (Figure 3d), respectively. Increases have been more obvious in the minimum temperature (Tmin) than in the other indicators, especially during 2000–2019 (Figure 3b).

The annual precipitation showed an increasing trend ($0.94\,mm/a$, $p > 0.10$, Figure 3e). The multi-year mean precipitation from 1957 to 2019 was $514.2 \pm 169.3\,mm$. Meanwhile, the relative humidity ($-0.03\,mm/a$) showed a decreasing trend (Figure 3f).

*4.2. Climate-Tree Growth Response Analysis*

The correlation analysis between the TRW chronology and the monthly meteorological factors showed that the radial growth of *Juniperus squamata* was more sensitive to temperature than to precipitation, in the Three-River Headwaters Region of the TP.

According to the results, the correlation between TRW chronology and Tmin data was significant at the 99% confidence level throughout all months, and was stronger than the correlations between TRW chronology and Tmax or Tmean. The correlation between TRW

chronology and Tmean was significant at the 95% confidence level in previous October and current March, significant at the 99% confidence level from July to September (previous and current year), from the previous November to current February, and in April and May of that current year. The correlation between TRW chronology and Tmax data was also significant at the 95% confidence level in previous November and current February, at the 99% confidence level in the previous December and January and July to September (previous and current year) (Figure 4a). In addition, the result of the climate response function analysis between chronology and temperature from July to September (previous and current year) was statistically significant.

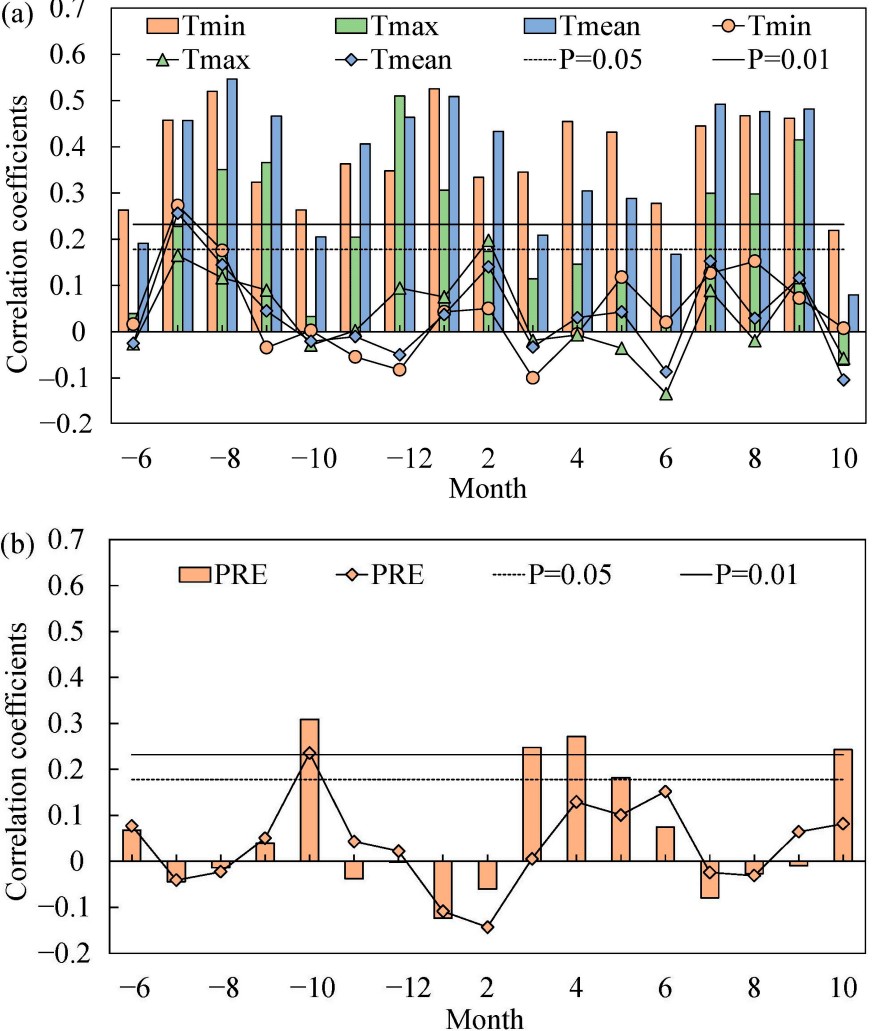

**Figure 4.** Results of correlation and response function analyses between TRW chronology of *Juniperus squamata* and (**a**) monthly temperatures (Tmin, minimum; Tmean, mean; and Tmax, maximum) and (**b**) precipitation (PRE) from prior June to current October for the period 1957–2019. The bar graphs denote the results of correlation analysis. The line graphs denote the results of response function analysis. The horizontal dashed and solid lines denote the 0.05 and 0.01 significance levels of the correlations, respectively. The "−" before the month denotes the previous year, For example, −6 denote June of the previous year.

In general, the correlation between TRW chronology and precipitation was weak. The TRW chronology was positively correlated with precipitation in spring months (March, April, May) and in October (previous and current year) at the 99% confidence level; the correlation between TRW chronology and precipitation was strongest in the previous year's October (Figure 4b).

*4.3. Annual Tmean Reconstruction*

Based on the ring width of *Juniperus squamata*, the TRW chronology was synthesized. According to the relevant results in the previous section, the standard chronology was significantly correlated with the mean temperature from July to September of the same year in the Three-River Headwaters Region of TP. It provided a basis to reconstruct the mean temperature of the growing season (July–September) in this region. Therefore, the current year chronology and the previous year chronology were used as independent variables in this study, and the mean temperature from July to September was reconstructed as the dependent variable to establish the regression equation, and the stability of the equation was tested using "leave-one-out" calibration.

We reconstructed historical (1905–2019) July to September Tmean data based on the relationship between TRW chronology and Tmean (1960–2019) in the Three-River Headwaters Region. The transfer function is as follows:

$$T_{7\sim9} = 10.638 + 0.536C_{STDx} + 1.129\ C_{STDx-1}.$$

In the equation, $T_{7-9}$ is the mean temperature series from July to September in the Three-River Headwaters Region, $C_{STDx}$ is the standard chronological series of the current year, and $C_{STDx-1}$ is the standard chronological series of the previous year

The climate reconstruction data's statistical parameters are listed in Table 1, and the results indicated that the reconstructions were reliable. For the reconstructed sequence (1960–2019), the correlation coefficient of the equation is 0.654, the explained variance 42.8%, and the corrected explained variance 40.8%, which indicates a good fit of the tree ring chronology to the mean temperature from July to September, which can explain the main variance of the measured sequence. F is the linear regression's parameter that is tested for statistical significance; the value of F in this study was 21.33, indicating that the data fit the regression model very well, and the confidence in the regression equation is very high. The results of the sign test and product average test for the reconstructed equation were statistically significant at the 1% significance level, indicating that the reconstructed sequence is consistent with the measured sequence, regardless of high or low frequency. The value of the reduced error was 0.246. It is generally believed that the detected data are reliable when the reduction error is positive. The product average T value is reasonable, indicating that the consistency between the two sequences is reliable. Therefore, the reconstruction dates back to 1905 and tracked well to both high- and low-frequency variations of the mean temperature from July to September (Figure 5a).

**Table 1.** Statistical parameters of mean temperature reconstruction process from July to September in the Three-River Headwaters Region of the Tibetan Plateau (1960–2019). r, correlation coefficient; $R^2$, explained variance; $R^2_{adj}$, explained variance after adjustment; F, regression test statistic; ST, sign test; $E_r$, reduced error; T, the product average; **, $p < 0.01$.

| r | $R^2$ | $R^2_{adj}$ | F | ST | $E_r$ | T |
|---|---|---|---|---|---|---|
| 0.654 | 0.428 | 0.408 | 21.33 | $30^+/30^-$ | 0.246 | 6.88 ** |

The comparison (1960–2019) between the observed temperature series (black line) and the reconstructed series (red line) showed that the TRW chronology better reflects the upward trend of temperature over the past 60 years (Figure 5a). However, the inter-annual variation between the reconstructed series and the observed series differed greatly in individual years (>1.5 °C in 1965, 1976, and 1994). The largest difference occurred in 1976 (the reconstructed value is 1.7 °C higher than the measured value). Further comparison of first-differences of observed and reconstructed temperatures revealed high-frequency consistencies, except for individual years (1965, 1975–1976, 1980, 1993–1994) (Figure 5b).

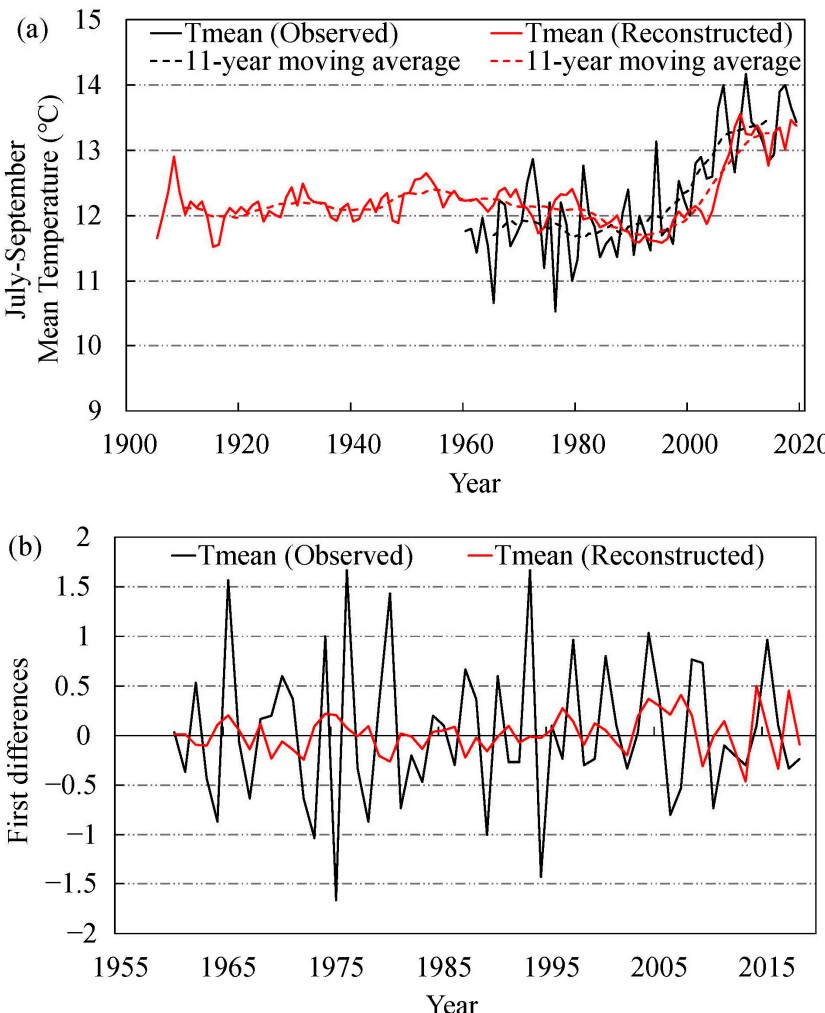

**Figure 5.** July–September mean temperature reconstruction in the Three-River Headwaters Region of the Tibetan Plateau. (**a**) Comparison between observed and reconstructed values of July–September mean temperature, (**b**) comparison between the first-difference data of observed and reconstructed July–September mean temperature.

From the July–September Tmean reconstruction series (1905 to 2019), the temperature change during this period can be roughly divided into two stages: from 1905 to 1999, when the temperature was within one standard deviation, and fluctuated around the average value,12.10 °C, and from 2000 to 2019, when the temperature showed a rapid rising trend and the growth rate was unprecedented over the past 95 years (Figure 5a).

In the 21st century, the radial growth of *Juniperus squamata* has accelerated dramatically as temperatures have risen, and the levels of increase were unprecedented in the last 95 years (Figure 6). The correlation results showed that the TRW chronology in the study area was closely related to the Tmean variation, reaching a 95% confidence level in 2008. In addition, the moving correlation results between the chronology and Tmean data gradually increased during the investigated period (1960–2019) (Figure 7). The results showed that the radial growth of *Juniperus squamata* was correlated with mean temperature.

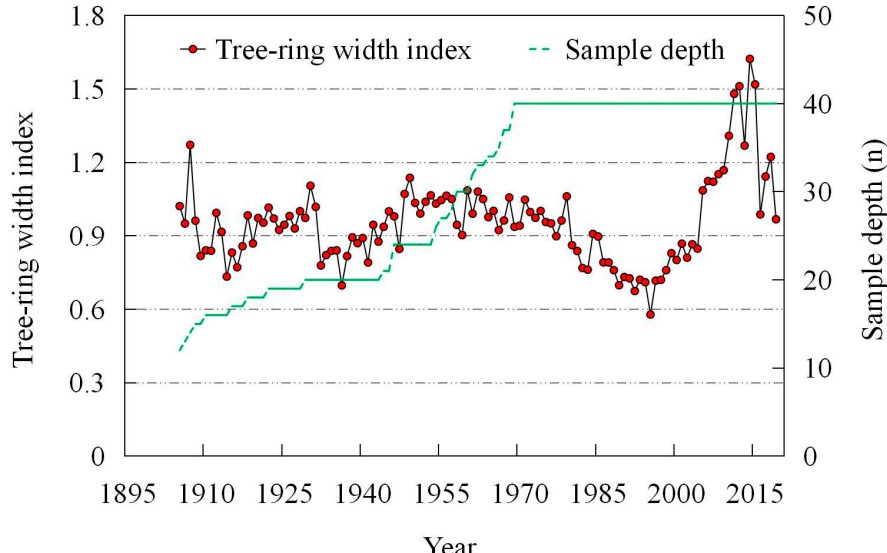

**Figure 6.** Standard TRW chronology and number of the discs sampled from *Juniperus squamata* in the Three-River Headwaters Region of the Tibetan Plateau.

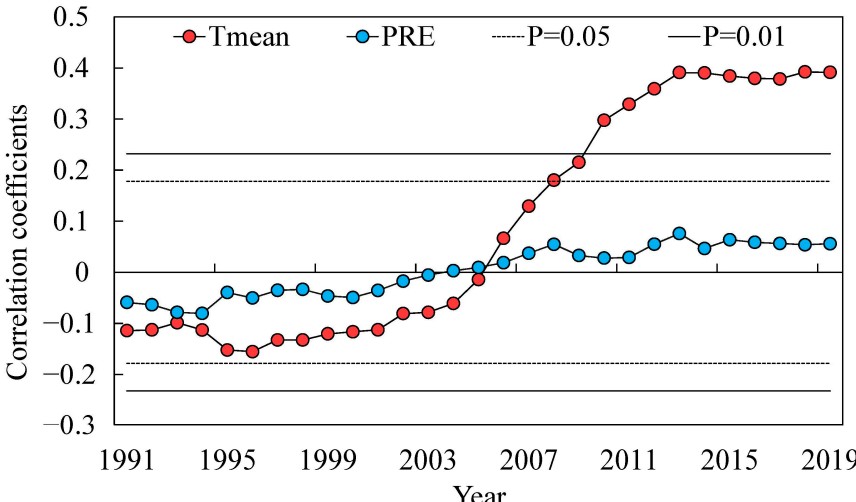

**Figure 7.** Results of moving correlation analysis between TRW chronology and annual mean temperature (Tmean) and precipitation (PRE) (using 34 years moving interval). The horizontal dashed and solid lines denote 95% and 99% confidence interval, respectively.

## 5. Discussion

### 5.1. Chronology Statistics Characteristics

We established a standard TRW chronology of *Juniperus squamata*, using the sample size with a subsample signal strength (SSS > 0.85), as the starting point of the chronology (Figure 6). The chronology can reflect the inter-annual fluctuation characteristics of tree radial growth. The standard deviation (SD) value was 0.31, which contains more climate information [56]. The first-order autocorrelation (AR1) value was 0.69, indicating that the growth was strongly affected by the previous year, which meant a relatively strong "legacy effect" [57,58]. The mean inter-series correlation (Rbt) value is 0.24, indicating that the change of TRW between different discs is consistent. The expressed population signal (EPS) value was 0.89, higher than the "0.85" threshold value, indicating that the samples could represent the local environmental characteristics [59,60]. The signal-to-noise ratio (SNR) value was 7.96, higher than the "4" threshold value, indicating that the samples contain abundant climate signals [61]. The statistical characteristics indicated that the TRW data contained reliable climate information (Table 2).

**Table 2.** Statistics of the standard TRW chronology of *Juniperus squamata* in the Three-River Head-waters Region of the Tibetan Plateau. SD, standard deviation; AR1, first-order autocorrelation; Rbt, mean inter-series correlation; SNR, signal-to-noise ratio; EPS, express population signal.

|  | Location | Elevation(m) | Time Length | Samples(n) | Mean Value |
|---|---|---|---|---|---|
| Type | 96.25° E 31.56° N | 4309.0 | 1957–2019 | 40 | 0.93 |
|  | SD | AR1 | Rbt | SNR | EPS |
| *Juniperus squamata* | 0.31 | 0.69 | 0.24 | 7.96 | 0.89 |

*5.2. Response to Temperature*

The correlation analysis between the TRW chronology and the monthly meteorological factors showed that the radial growth of *Juniperus squamata* was more sensitive to temperature than to precipitation in the Three-River Headwaters Region of the TP. Moreover, the radial growth of *Juniperus squamata* was positively correlated with the temperature in most months (June of last year–October of current year), especially the monthly mean minimum temperature (Figure 4a).

The chronology was significantly positively correlated with the mean temperature from July to September. Research shows in some alpine and humid regions, temperature rise is conducive to the acceleration of the radial growth and the extension of the growing season, which increases forest productivity [35]. This is because temperature rise during the growing season (July to September) can promote the synthesis and transport of auxin, and the division of cambium cells, as well as the accumulation of carbohydrates. Some studies have pointed out the mean temperature threshold for the start time of tree cambium cells activity is 6–8 °C, In cold and humid regions, the end time of radial growth of trees is mainly controlled by temperature [35,61–63]. Therefore, the critical period of growth and development of *Juniperus squamata* in Nangqian County is mainly concentrated in May to September. The radial growth of *Juniperus squamata* was positively correlated to the mean temperature in March. A study of Smith fir trees in Sejila Mountain in southeastern Tibet showed that the lowest temperature threshold limiting the onset of wood partialization is $0.7 \pm 0.4$ °C [63–65], which is far lower than temperatures previously reported. Therefore, under the ideal state, the beginning of the growth period of the *Juniperus squamata* is around March (1.07 °C). There is a significant positive correlation between the radial growth of *Juniperus squamata* and temperature in December. *Juniperus squamata* is deep-rooted plant, driven by photosynthesis; its main nutrients come from the soil. In Nangqian County, the ground temperature in winter is higher than that in spring [66]. In spring, the ground temperature in most parts of the TP is still at a subzero state. Permafrost requires so much heat to melt, that its temperature cannot rise quickly. In winter, the ground freezes and releases heat, which makes its temperature decrease slowly and maintains a relatively high temperature [51,52,67,68]. This is a unique phenomenon in seasonal frozen soil or permafrost area of the TP. The surface temperature is mainly affected by atmospheric temperature. The radial growth of *Juniperus squamata* has a significant positive correlation with the maximum temperature from November of the previous year to February of the current year. Generally, the xylem is affected by the accumulated temperature in winter and spring, and the warm winter prevents the leaf tissue from freezing and maintains the normal metabolism of trees, which is beneficial to the next year tree growth [9,46,69,70]. Therefore, the higher the maximum temperature is, the more favorable to the xylem growth of *Juniperus squamata*, and the easier it is to form wide rings.

Based on the correlation between TRW chronology and climate variables, the annual Tmin was the key climatic factor influencing radial growth of tree in conifer species at higher elevations on the southeastern TP. The conclusion is consistent with other studies [15,16,37,71–73]. Critical low temperature limits xylem growth on the forest line in the southeastern TP, especially Tmin in July [3,65,74,75]. Temperature is the main factor restrict-

ing *Sabina przewalskii* on the northeastern margin of the Tibetan Plateau. The temperature of July and August is significantly positively correlated with its annual ring index [76–78]. The thermal index required for alpine shrubs at an altitude of 4000–4500 m is the annual average temperature of $-1\sim2$ °C, the average temperature of the hottest month is $7\sim10$ °C, the average temperature of the coldest month is $-11\sim-8$ °C, and the extreme minimum temperature is $-30\sim-8$ °C [9,35,56,65]. In Nangqian County, from 1957 to 2019, the mean temperature was 4.43 °C, the hottest mean monthly temperature was 25.16 °C, and the coldest mean monthly temperature was $-19.80$ °C. Among the three indicators, both the annual mean temperature and the hottest mean monthly temperature met the thermal index. Clearly, the minimum temperature was the main factor restricting growth, which is consistent with other results.

### 5.3. Response to Precipitation

The correlation analysis between the TRW chronology and the monthly meteorological factors showed that the radial growth of *Juniperus squamata* had a weak correlation with precipitation in the Three-River Headwaters Region of the TP (Figure 4b).

The radial growth of *Juniperus squamata* is positively correlated with the precipitation from March to May, indicating that its increases make the division of cambium cells more vigorous. Moreover, the precipitation and humid air environment in early spring are conducive to the budding of leaves [79–81]. The radial growth of *Juniperus squamata* is positively correlated with the precipitation in October because the cambium cells is still active at the end of the growing season, and rainwater can promote photosynthesis and accumulate nutrients. Moreover, precipitation makes the soil hydration sufficient, and the *Juniperus squamata* does not need to use the water in the body, so fallen leaves are reduced [82].

### 5.4. Fluctuation Characteristics of the Reconstruction Series

The reconstructed series reflects the temperature fluctuations from July to September in past 115 years in the Three-River headwaters region, which can be roughly divided into two stages (Figure 5a). From 1905 to 1999, the mean temperature was within one standard deviation, and fluctuated around the average value; this is similar to the trend of mean maximum temperature from May to June between 1360 and 2005 with *Sabina Tibetica* tree rings reconstructed by Shi et al. in Zaduo, Qinghai Province [83]. From 2000 to 2019, the temperature showed a rapid rising trend, with a growth rate that was unprecedented in the past 95 years. Kyimu et al. found that the minimum temperature has also been increasing in the past two decades when the minimum temperature was reconstructed with tree rings in the southeast of the Tibetan Plateau [15]. The explained variance of our July–September mean temperature reconstruction (42.8% of actual temperature) is comparable to, or even higher than, previous climate reconstructions of the TP and nearby regions [46,51,52]. In the 1980s, 1987 was a turning point, as the plateau monsoon had reached its peak in 1984. With stronger monsoons, temperature increases, and there is a three-year lag in its impact on summer and autumn temperatures. There are significant differences in the temperature distribution in the region around 1987 [51,79,84–86], and this point in time is consistent with our study. From 1987 to 1997 in the Three-River Headwaters Region, there were 7 years with long durations and wide ranges of snow cover, specifically in 1987, 1989, 1990, 1993, 1994, 1995 and 1997 [28,74,77,87]. Therefore, after 1987, there was a 7-year cold period. After 1998, the increasing rate of summer temperatures has accelerated significantly [88,89]. This year was in the late-1990s, which was an important turning point (Figure 5a). The average temperature from July to September could not represent the complete summer or autumn. The growth of grassland is significantly correlated with temperature in the same period, and the maximum forage yield in the region is in the months of July and August [75]. Since 2006, the forage yield of different grassland types has shown an increasing trend, and vegetation is getting better, which has a good correspondence with the sequence of

this study. The temperature series shows that after 2006, the climate has entered to a warm period.

Further comparison of first-differences of observed and reconstructed temperatures revealed high-frequency consistencies, except for individual years (1965, 1975–1976, 1980, 1993–1994). Although there was some difference in the first-differences between the reconstruction value and the observed value in these years (>1.5 °C in 1976, 1980, and 1993), the largest difference occurred in 1975 (the reconstruction value was 1.8 °C higher than the observed value), but the difference was within the acceptable range (Figure 5b). This is similar to the first-differences analysis performed by Panthi et al. who described retrospective winter temperatures based on the ring width of *Rhododendron campanulatum* D.Don shrubs in the central Himalaya [90]. Considering that tree growth is affected by a variety of external factors, and temperature from July to September is but one of them, it cannot be ruled out that trees are impacted by other factors in individual years. Generally, however, the reconstructed series change trend fit with the observed temperature (from July to September). Furthermore, it was shown that the shrub *Juniperus squamata* was closely correlated with temperature and had good growth potential.

## 6. Conclusions

In order to investigate the effects of climate change on alpine shrubs, we have established a standard chronology of *Juniperus squamata* tree-ring width in the Three-River Headwaters Region of the Tibetan Plateau by using a dendroclimatology method, and its response to climate change was analyzed. Based on the relationship between temperature and tree growth, the mean temperature from July to September in the past 115 years was reconstructed. There were three primary findings.

(1) Meteorological data show that the temperature in Nangqian County of the Tibetan Plateau has increased continuously in the past 60 years, and the minimum temperature has the most significant change (0.63 °C/10a), especially between 2000–2019 (0.12 °C/a), while the precipitation has not changed significantly (0.94 mm/a, $p > 0.10$). The changes in our reconstructed temperature series can be roughly divided into two stages: from 1905 to 1999, when the temperature fluctuated around the average value, 12.10 °C, and from 2000 to 2019, when the temperature showed a significant upward trend (0.08 °C/a). The similarity and verification results between the current reconstructed series and the previously reported historical temperature records suggests reliability of the new reconstructed series.

(2) During 1957–2019, the response of radial growth of *Juniperus squamata* to temperature was significantly stronger than that of precipitation, especially in the last 20 years; the radial growth of *Juniperus squamata* was positively correlated with temperature ($p < 0.01$). Compared to the maximum temperature and mean temperature, the correlation between radial growth of *Juniperus squamata* and minimum temperature was more significant.

(3) According to our results, radial growth of high altitude shrubs in the study area kept pace with rising temperature, which could be divided into two stages: a steady increase trend from 1905–1999; a significant increase from 2000–2019, where temperature entered a rapid growth period.

**Author Contributions:** Conceptualization, G.Z. and Z.X.; methodology, Z.L.; software, M.K.; validation, Z.X., Z.L. and M.K.; formal analysis, Z.X.; investigation, J.L. and Y.H.; resources, J.L. and Y.H.; data curation, G.Z.; writing—original draft preparation, G.Z.; writing—review and editing, Z.X., Z.L. and M.K.; visualization, G.Z.; supervision, G.Z.; project administration, G.Z.; funding acquisition, Z.X. All authors have read and agreed to the published version of the manuscript.

**Funding:** This research was funded by the Second Tibetan Plateau Scientific Expedition and Research Program (STEP), (2019QZKK0608), the National Natural Science Foundation of China (42177319), and the Beijing Municipal Education Commission for Inter-disciplinary Program "Ecological Restoration Engineering".

**Data Availability Statement:** Not applicable.

**Conflicts of Interest:** The authors declare no conflict of interest.

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
