# Peer review of "The Radial Growth of Juniperus squamata Showed Sharp Increase in Response to Climate Warming on the Three-River Headwaters Region of Tibetan Plateau since the Early 21st Century"

_forests, doi:10.3390/f14061219_

Round 1

Reviewer 1 Report

Dear collaueges, 

I read Your manuscript with pleasure, but I have got some remarks:

1. Sabina squamata is synonym of Juniperus squamata Buch.-Ham. ex D. Don (see http://www.theplantlist.org/tpl1.1/record/kew-2484374). Also, You should specify the authority of plant species. 

2. In 3.1 Section the number of discs was mentioned, but not a number of measured radii. 

3. In 3.3. Section You pointed out the response function analysis. Nevertheless, below in the manuscript the results of correlation analysis only are discussed. It is especially incomprehensible because You used regression analysis, and response function is useful instrument for choosing the better predictors for regression. 

4. There is a similar problem with detrending. In 3.1. Section You pointed out the trend removal by neg-exponential or linear regression. Do You use raw or detrended standard chronology for regression analysis in the sections below? 

5. The footnotes of Figure 3 and Table 2 must be more detailed. E.g., it is hard to understand for readers what does the letters on Fig. 3 means. 

6. In line 282 there is an equation for temperature (T7-9) reconstruction. In context, T7-9 is the minimal temperatures. Nevertheless, in line 283 mentioned than this is the mean temperature. Is the mean minimum for these months? Moreover, below (e.g., line 307) this weather characteristic indicated as Tmean. The designations must be unified through the text. 

7. You should proofread the manuscript: 

- there is a duplicated part of text (l.170–173 and l. 230–232); 

- there are some offensive terms (e.g., in l. 83 'above the line' instead of 'above the tree line', in l. 339 'division of cambium' instead of 'division of cambium cells'); 

- there are some misspels (e.g., 'september' in l. 274 or '--' instead of '–' in l. 382). 

Best regards! 

Reviewer 2 Report

In the manuscript “The radial growth of Sabina squamata showed sharp increase in response to climate warming on the three-river headwaters region of Tibetan Plateau since the early 21th centaury” Guoqing and co-authors use a chronology of a shrub species to perform a reconstruction of the summer temperature of the area for the last 115 years. Reported results show a sudden increase of the tree-ring growth since the beginning of the 21st century in accordance with the increase of the temperature recorded by the nearest meteorological station. In my opinion, the manuscript is in line with the aims of the journal Forests and its special issue Responses of Forest Ecosystems to Climate Change in section Forest Meteorology and Climate Change reporting original chronologies of a species not intensively investigated. However, in the present form, the work is characterized by methodological flaws that may drive the authors to erroneous conclusions. Major and minor comments are reported in the attached file, line number is referred to the line reported on the document that can be downloaded from the Journal.

The manuscript will benefits from a English revision. 

Reviewer 3 Report

The manuscript submitted for review is well prepared, following classical procedures used in dendrochronology. The authors showed stronger relationships with temperature than with precipitation. With temperature, the strong influence of Tmin is not in doubt and is justified under these climatic conditions. Consequently, future reconstruction of Tmin should be considered, especially since the period studied falls before the rapid warming.

I congratulate the authors on a well-prepared work.

Method needs to be clarified. The sentence “More than 40 shrub discs were selected away from human activities” is unclear. How was the material taken? Were they random trees or were they selected in some way? Did each disk come from a separate tree? How many directions were analyzed in each disk?

It seems as if the manuscript was previously prepared for another journal and no changes were made. The names should be replaced with numbers, the formatting of citations should be changed to MDPI, the letter [J] - probably a remnant from another journal, the use of full words Figure instead of "fig." etc.

Since the results of (4.2) contain elements of discussion, please consider including them in the discussion or compiling the two chapters into one.

l. 228 there is “ARI” should be “AR1”

Table 1 should be modified, the names of the statistics (SD, etc.) are also the header

The resolution of the figures should be better.
